# Intergroup attitudes and contact between Spanish and immigrant-background adolescents using network analysis

**María Sánchez-Castelló** [1,2]*, **Marisol Navas**[1,2], **Antonio J. Rojas**[1,2]

1 Department of Psychology, Faculty of Psychology, University of Almería, Almería, Spain, 2 Center for the Study of Migration and Intercultural Relations (CEMyRI), Almería, Spain

* msc943@ual.es

**Data Availability Statement:** The database used in this research has been made publicly available and can be accessed at Open Science Framework

## Abstract

This study aimed to analyze the relationship among different evaluative reactions of the intergroup attitudes and contact in Spanish adolescents evaluating different ethnic minorities and in immigrant-background adolescents evaluating Spanish youth. This study was based on psychosocial models of great impact in the study of intergroup relations such as the Stereotype Content Model and the Behaviors from Intergroup Affect and Stereotypes Map, and incorporated a new approach to the study of attitudes: psychological networks. In total, 1122 Spanish adolescents and 683 adolescents with an immigrant background (Moroccan, Romanian or Ecuadorian origin) participated in the study, aged from 12 to 19 years. They answered a questionnaire with measures of stereotype dimensions (morality, immorality, sociability and competence), emotions (positives and negative), behavioral tendencies (facilitation and harm) and contact (quantity and quality). The results show similar structural patterns in the six studied groups, with emotions acting as links between stereotypes and behavioral tendencies. Moreover, positive and negative stereotype dimensions appeared as independent dimensions that were part of different processes: sociability and morality, and competence to a lesser extent, were related to facilitation behaviors through positive emotions, while immorality was related to harm behaviors through negative emotions. This could indicate that, to achieve successful intergroup relations involving cooperation and the development of friendly relationships, it would be appropriate to intervene in parallel in these two pathways. Due to the centrality of positive emotions (and sociability and immorality) and, therefore, their capacity to affect the entire network, focusing interventions on these variables could be an appropriate strategy to achieve overall positive attitudes.

## Introduction

The extensive literature of intergroup attitudes shows the importance of its study in contemporary societies due to the great impact that these attitudes have on intergroup relations. The increase in diversity in current societies is reflected in the presence of students with an immigrant background in secondary schools (in Spain, about 10% of the total student body during the 2019–2020 academic year whereas this percentage was less than 2% in the 2000–2001

(OSF): https://osf.io/reyh3/?view_only=
f7bf63a972c14a4bb0bfe97c6c05e7c9.

**Funding:** This study is part of the project
'Prejudiced attitudes, acculturation process and
adjustment of immigrant and host adolescents'
[Reference PS2016-80123-P], funded by the
Ministry of Economy, Industry and
Competitiveness (Spain). The funders had no role
in study design, data collection and analysis,
decision to publish, or preparation of the
manuscript.

**Competing interests:** The authors have declared
that no competing interests exist.

academic year, according to the Spanish Ministry of Education, Culture and Sport [1, 2]).
These spaces, which are the main contexts of socialization for adolescents, are also the main
scenarios where intergroup contact takes place. In addition, adolescence is a key developmental stage in the formation of intergroup attitudes, in which interventions to improve them can
be particularly effective [3].

In the present study, we analyzed the structure of intergroup attitudes of different groups
of adolescents living in Spain from both the majority and minority perspective, and their relationship with the quality and quantity of contact. For this purpose, this study was based on
psychosocial models of great impact in the study of intergroup relations such as the Stereotype
Content Model (SCM) [4] and the Behaviors from Intergroup Affect and Stereotypes Map
(BIAS Map) [5, 6], and incorporated a new approach to the study of attitudes: psychological
networks [7].

## Current perspectives in the study of intergroup attitudes

Many conceptualizations have described attitudes as evaluations about attitudinal objects
(people, groups, things, ideas, etc.), ranging from positive to negative, that can be inferred
through beliefs, affect and behavior [8, 9]. According to SCM [4], group representations (stereotypes) are structured in two fundamental evaluative dimensions: warmth and competence.
The first one refers to the perceived intentions of the out-group or their members with aspects
related to kindness and goodness (e.g., whether they are sincere, affectionate), and the second
one refers to the capacity of out-group or their members to achieve their objectives or intentions (e.g., whether they are intelligent, efficient). This model also shows that different combinations in the warmth and competence dimensions cause different patterns of affective
reactions (i.e., contempt, admiration, compassion and envy).

Subsequent studies have shown three stereotype dimensions, dividing in two the dimension
of warmth: morality and sociability [10, 11]. The morality dimension would include characteristics related to the correctness of out-group targets (e.g., honest, trustworthy), while sociability
would include characteristics related to cooperation and connection with others (e.g., friendly,
kind [12]). Other authors [13, 14] have included the negative pole of morality (i.e., immorality)
and have shown its importance in out-group evaluations. Previous research [13–17] has
shown evidence of the theoretical importance and the diagnostic capacity of the negative pole
of the morality domain (i.e., immorality) in the out-group's impression formation (i.e., for this
dimension, negative information is more enlightening than positive information).

The extension of SCM [4], i.e. the BIAS Map [5, 6], provides a general structure about how
the stereotypes and emotions experienced towards out-groups are related to behaviors towards
them. Intergroup emotions are those that people experience due to belonging to a group and
that result from self-categorization and salient social identity [18]. Studies based on the BIAS
Map [5, 6] as well as in the Intergroup Emotion Theory (IET; [19]) have showed the powerful
effects of intergroup emotions acting as mediators between stereotypes and behavioral tendencies. The meta-analysis of Talaska et al. [20] showed that emotions towards ethnic minorities
were more direct predictors of discriminatory behaviors than beliefs and stereotypes. The IET
model, however, qualifies that the sequence *cognition*, *emotion* and *behavior* is not always unidirectional, because emotion has broad forward and backward effects on cognition (i.e.,
"appraisals" and emotions influence each other) [19].

The BIAS Map hypothesis have been tested in different cultural contexts. In Norway, Bye
and Herrebrøden [21] found support general for the BIAS Map predictions, with warmth and
competence as stereotype dimensions, but with some variations. In Spain, Cuadrado et al. [22]
showed the relationship between stereotypes (morality, sociability and competence

dimensions) and behavioral tendencies of facilitation through positive emotions in adults and adolescents, from the perspective of the majority and minority. SCM extended (three stereotype dimensions) and BIAS Map have also been used as a theoretical framework for the development of two studies on the adolescent population in Spain [23, 24]. In the first study [23], morality predicted facilitation and harm intentions (through positive and negative emotions) and harm intentions (through negative emotions), while sociability predicted only facilitation intentions through positive emotions. In the second study [24], morality predicted the intentions of facilitation and harm through emotions of admiration and contempt, respectively. In general, morality seems to be a key determinant in predicting intergroup behavior through emotions. Some of the above findings have been captured in the three key assumptions of the Moral Primacy Model [25]: 1) Morality, sociability, and competence are distinct dimensions that make unique contributions to impression formation; 2) Morality has a primary role in the formation of impressions and the evaluations we make about other people and groups both at different stages of impression formation and in behavioral outcomes; 3) The primary role of morality is due to its close link to the judgment of whether other social targets represent an opportunity or a threat.

Some researchers consider both positive and negative attitudes to be separate dimensions and point to the appropriateness of including both dimensions in the study of attitudes [26, 27]. Pittinsky et al. [28] showed that positive and negative attitudes are functionally separable since they relate to behaviors according to their valence (i.e., positive attitudes are more strongly linked to positive behaviors, while negative attitudes are more strongly linked to negative behaviors). Therefore, improving intergroup relations may involve both the reduction of negative intergroup attitudes and the promotion of positive intergroup attitudes, because this reduction and promotion may be linked to different psychological processes [27].

Consistent with the SCM [4] and the BIAS Map [5, 6], stereotypes, emotions and behavioral tendencies were used as indicators of intergroup attitudes in this study. The present study also includes the stereotype dimension of immorality in order to test whether the positive (i.e., morality, sociability or competence) and negative (i.e., immorality) poles are related to positive and negative emotions and to positive (facilitation) and negative (harm) behaviors, respectively.

## The study of attitudes from a network approach

Of the many frameworks used in the study of attitudes, recently, a new approach is gaining significant interest: the psychological networks [7, 29–31]. This approach has led to a new way of looking at the concept of attitude and a new measurement model to apply to empirical data [7]. Traditionally, for the study of psychological constructs, it is assumed that there are underlying latent variables that explain the scores of the observable variables (e.g., self-reported items) and the correlations between these observable variables. From a network approach, instead of considering observable variables as indicators of a latent variable, they are considered as autonomous entities related in a network of dynamic systems [32]. This allows us to overcome some of the conceptual and methodological limitations or problems found in research based on latent variable models (and derived from the tripartite attitude model in attitudes research; for a more detailed view see [7, 31]). In addition, networks improve other methodological approaches such as the multiple regression model or factor analysis, fundamentally because it shows in a single plot the relationships between variables of large datasets, building a network and providing an easier interpretation [33].

Within this framework, a new measurement model has been developed to explain how the different information bases of attitude relate to the attitude construct and to explain

correlations between them: the Causal Attitude Network (CAN) model [7]. This model conceptualizes attitudes as networks consisting of evaluative reactions (beliefs, feelings and behaviors) toward the attitude object and interactions between these reactions. The present study followed this line and, therefore, no directionality was established between the different evaluative responses, to be compatible with the bidirectionality between cognition and emotions defended by some theories of emotions such as the IET [19] and found in some studies in recent research [22].

## Intergroup attitudes and contact

One of the most important approaches to improving intergroup relations is intergroup contact theory, based on the contact hypothesis developed by Allport [34] (see also the meta-analysis of Pettigrew and Tropp, [35]). This hypothesis proposes that intergroup contact could decrease prejudice and improve intergroup relations between members of majority and minority groups (under appropriate conditions: equal status between the groups, common goals, intergroup cooperation, and the support of authorities, law, or custom). However, in Pettigrew and Tropp's meta-analysis, it was observed that these conditions were not necessary for a reduction in prejudice to occur and these authors proposed that the simple familiarity that occurs with contact generates liking. In addition, they point out that the relationship between familiarity and liking could be influenced by variables such as the reduction of uncertainty, intergroup anxiety and threat. The study by Laurence et al. [36] showed that increased out-group size was related to intergroup attitudes towards them only in more segregated communities (but not in the case of more integrated communities), and that this is explained by an increase in perceived threat and a decrease in the quality of contact (when the increase in out-group size exceeds a certain value).

Specifically, it is the quality of contact rather than the quantity of contact that has shown the greatest effect on intergroup attitudes [37]. Quality of contact refers to the participant's subjective sense of the nature of the contact experience [38]. In a study by Binder et al. [39], both the quality and quantity of contact influenced over time two indicators of prejudice: the desire for social distance and negative intergroup emotions, with the quality of contact being the most relevant variable. Some studies have shown that positive contact, conceptualized as cross-group friendship, is related to different measures of intergroup attitudes, especially emotions like admiration and sympathy [37, 38]. Bobowik et al. [40] conducted a study using a social network approach in Spain with adults of immigrant background and found that the proportion of intergroup contacts (with native-born population) representing strong ties (i.e., close relationships such as friends, romantic partners, and immediate and extended family members) and ethnic diversity among these ties were associated with more favorable attitudes toward outgroups. In another study conducted in the same community with 15- and 16-year-old students, it was observed that students who reported frequent interaction with other students of different nationalities and religions held more positive intergroup attitudes and the presence of a greater number of immigrants in classrooms was related to lower levels of xenophobia [41].

## The present study

This study aimed to examine the relationship between different evaluative reactions of the attitude (stereotypes, emotions, behavioral tendencies) and contact in different groups of adolescents in Spain: in the majority group (Spanish adolescents evaluating immigrant youths from different origins) and in three minority groups with an immigrant background (adolescents evaluating Spanish youth from three of the main immigrant groups in Spain according to the

Spanish National Institute of Statistics [42]: Moroccans, Romanians and Ecuadorians). The application of network models in different groups and/or towards different targets will allow us to explore the structure and dynamic properties of intergroup attitudes (i.e., the relations between different evaluative reactions). As far as we know, the present study is one of few studies that simultaneously included the positive and negative pole of stereotypes, emotions and behavioral tendencies applied to the adolescent population (majority and different ethnocultural minorities) using a network approach.

Firstly, we think morality, and especially immorality, are probably the most influential stereotypical dimensions in the network (Hypothesis 1). We also expect to find a central role for emotions within the network since the literature has shown their powerful effect acting as the link between all the other information bases of the attitude in adults [5, 6, 19, 22] and adolescent population [23, 24] (Hypothesis 2). Thirdly, we also hypothesize that the relationship between stereotypes and behavior through emotions is a function of their valence (Hypothesis 3). That is, we expect to find a relationship of stereotypes to facilitation tendencies through positive emotions and a relationship of stereotypes to harm tendencies through negative emotions [26–28]. Finally, we presume that the quality of contact has more influence on the network than the quantity of the contact, and that it is related to the rest of the network, mainly through emotions [37, 38] (Hypothesis 4).

## Materials and methods

### Participants

A sample of 1122 Spanish adolescents and 683 adolescents with an immigrant background from 12 to 19 years voluntarily participated in this study. From the sample of Spanish adolescents, 488 evaluated youth of Moroccan origin (SM group: $M_{age}$ = 14.79, $SD_{age}$ = 1.23; 52.4% girls), 314 evaluated youth of Romanian origin (SR group: $M_{age}$ = 14.84, $SD_{age}$ = 1.53; 54.2% girls) and 320 evaluated youth of Ecuadorian origin (SE group: $M_{age}$ = 15.36, $SD_{age}$ = 1.57; 51.7% girls). From the sample of adolescents with an immigrant background, 360 were of Moroccan origin (M group: $M_{age}$ = 15.16, $SD_{age}$ = 1.36; 58.7% girls), 137 were of Romanian origin (R group: $M_{age}$ = 15.04, $SD_{age}$ = 1.45; 56.6% girls) and 186 were of Ecuadorian origin (E group: $M_{age}$ = 15.27, $SD_{age}$ = 1.42; 48.91% girls). All of them evaluated Spanish youth. They were enrolled in 17 public secondary schools in different municipalities in five Spanish provinces (Almería, Alicante, Castellón, Madrid and Murcia). The valid percentages of first-generation adolescents were 63.2%, 68.9% and 41.6% for adolescents of Moroccan, Ecuadorian and Romanian origin, respectively. For second generation, the valid percentages were 36.8%, 31.1% and 58.4% for adolescents of Moroccan, Ecuadorian and Romanian origin, respectively. The mean age of arrival in Spain was 4.82 years old ($SD$ = 4.39) for Moroccan origin adolescents, 5.49 ($SD$ = 4.35) for Romanian origin adolescents, and 6.38 ($SD$ = 4.83) for Ecuadorian origin adolescents. Average perceived socioeconomic status score (ranging from 1 to 10 points) was 6.38 ($SD$ = 1.36) for SM group, 6.39 ($SD$ = 1.29) for SR group, 6.43 ($SD$ = 1.33) for SE group, 6.14 ($SD$ = 1.61) for M group, 6.36 ($SD$ = 1.28) for R group, and 6.69 ($SD$ = 1.36) for E group.

### Variables and instruments

Participants answered a questionnaire with two similar versions, changing the out-group evaluated. Spanish adolescents answered the questionnaire assessing Moroccan youth, Romanian youth or Ecuadorian youth, and adolescents with an immigrant background assessed Spanish youth. Questionnaires contained instruments to measure the following variables. The estimated reliability coefficients (Cronbach's alpha) are included in Table 1.

**Table 1. Coefficients of estimated reliability (Cronbach's alpha and Split-half after Spearman-Brown correction) of all the variables in the six sub-samples.**

| Variables | SM group α | SM group Split-half | SR group α | SR group Split-half | SE group α | SE group Split-half | M group α | M group Split-half | R group α | R group Split-half | E group α | E group Split-half |
|---|---|---|---|---|---|---|---|---|---|---|---|---|
| Morality | .82 | .84 | .79 | .80 | .80 | .80 | .65 | .65 | .64 | .69 | .66 | .72 |
| Immorality | .80 | .74 | .86 | .84 | .84 | .81 | .77 | .74 | .82 | .81 | .80 | .78 |
| Sociability | .85 | .86 | .85 | .87 | .81 | .80 | .72 | .71 | .75 | .80 | .82 | .80 |
| Competence | .76 | .76 | .77 | .77 | .78 | .76 | .68 | .63 | .69 | .66 | .69 | .67 |
| Positive emotions | .86 | .86 | .85 | .86 | .87 | .87 | .77 | .79 | .80 | .77 | .81 | .83 |
| Negative emotions | .83 | .81 | .86 | .86 | .79 | .75 | .72 | .71 | .81 | .79 | .81 | .77 |
| Facilitation | .91 | .89 | .91 | .87 | .93 | .91 | .84 | .77 | .85 | .82 | .85 | .79 |
| Harm | .79 | .82 | .85 | .88 | .72 | .77 | .67 | .67 | .77 | .85 | .60 | .72 |
| Quantity of contact | - | - | - | - | - | - | - | - | - | - | - | - |
| Quality of contact | .79 | - | .73 | - | .82 | - | .73 | - | .74 | - | .81 | - |

SM group = Spanish adolescents who evaluate Moroccan-origin youth; SR group = Spanish adolescents who evaluate Romanian-origin youth; SE group = Spanish adolescents who evaluate Ecuadorian-origin youth; M group = Moroccan-origin adolescents; R group = Romanian-origin adolescents; E group = Ecuadorian-origin adolescents.

**Stereotypes.** This variable was measured through a 17-item scale elaborated from López-Rodríguez et al. [43] and Sayans-Jiménez et al. [14]. The scale consisted of four subdimensions: morality (4 items: honest, trustworthy, sincere and respectful), immorality (5 items: aggressive, malicious, harmful, treacherous and false), sociability (4 items: friendly, warm, likeable and kind) and competence (4 items: intelligent, skillful, competent, and efficient). Participants were asked to what extent each adjective described to out-group youth (Spaniards, Moroccans, Romanians or Ecuadorians). The response-scale ranged from 1 (*not at all*) to 5 (*very much*). Average scores were calculated (ranging from 1 to 5), so that higher scores indicated greater perception of morality, immorality, sociability and competence of the out-group evaluated.

**Intergroup emotions.** This variable was measured through a 13-item scale elaborated from Cuadrado et al. [44] and the subscale of emotions of the Prejudiced Attitude Test [45]. The scale consisted of two subdimensions: positive emotions (admiration, respect, security, understanding, love and sympathy) and negative emotions (mistrust, indifference, hate, anger, fear, disgust and discomfort). Participants were asked to what extent they feel each emotion toward the out-group. The response-scale ranged from 1 (*nothing*) to 5 (*a lot*). Average scores were calculated (ranging from 1 to 5), so that higher scores indicated more intense emotions (positive or negative) toward the out-group evaluated.

**Behavioral tendencies.** This variable was measured through a 12-item scale from López-Rodríguez et al. [46], based on the BIAS Map [5, 6]. The scale consisted of two subdimensions: facilitation (caring for them, helping them, protecting them, cooperating with them, teaming up with them, joining them) and harm (excluding them, ignoring them, going beyond them, insulting them, assaulting them, harassing them). Participants were asked to what extent they would be willing to do each behavior toward out-group people. The response-scale ranged from 1 (*nothing*) to 5 (*a lot*). Average scores were calculated (ranging from 1 to 5), so higher scores indicated greater willingness to engage in facilitation and harm behaviors towards the out-group evaluated.

**Quantity of contact.** A question (In general, how much contact have you had or do you have with -the out-group-?) was used as an indicator of the amount of contact. The response-scale ranged from 1 (*nothing*) to 5 (*a lot*). Higher scores indicated a higher quantity of contact.

**Quality of contact.**   This was measured using a 3-item scale based on Islam and Hewstone [47]. The participants were asked about the quality of the contact with the out-group by using a list of bipolar adjectives with three response alternatives ranging from unpleasant-pleasant, involuntary-voluntary and superficial-intimate. Average scores were calculated (ranging from 1 to 5), so higher scores indicated better quality of contact.

**Socio-demographic variables.**   Participants reported their sex, age, place of birth, age when they arrived to Spain, countries of birth of their parents and perceived socioeconomic status.

## Procedure

After carrying out a residential census analysis, five sampling zones (Almería, Alicante, Castellón, Madrid and Murcia) were chosen with the largest immigrant population [34]. Specifically, the percentages of foreign population in these provinces were 21.37%, 19.84%, 15.18%, 14.15% and 14.64%, respectively according to the National Institute of Statistics [48]. Members of the research team contacted the public secondary schools in these areas to choose those with the greatest proportion of immigrant students. All the permits required from the local governments and schools were obtained. Once the school authorities agreed to participate, the adolescents' parents were informed of the relevant aspects of the research (voluntary and anonymous participation, use of the data for scientific purposes, etc.) and they signed a consent form. The questionnaires were administered by members of the research team in the classrooms, in paper-and-pencil format. The duration was approximately 30 minutes. The study was approved by the Bioethics Committee in Human Research of the University of the authors. The database used in this research has been made publicly available and can be accessed at Open Science Framework (OSF): https://osf.io/reyh3/?view_only=f7bf63a972c14a4bb0bfe97 c6c05e7c9.

## Results

### Data analysis

We computed the descriptive statistics, partial correlations and Cronbach alpha coefficients for each scale and group using IBM SPSS Statistics v.25. We estimated an undirected psychological network where each node represents one of the ten domains measured: four stereotype dimensions (morality, immorality, sociability and competence), two intergroup emotions (positive and negative emotions), two behavioral tendencies (facilitation and harm) and quantity of contact and quality of contact. We used these 10 domain scores as variables in the Gaussian Graphical Model (GGM; [49]), a regularized partial correlation network [50]. Extended Bayesian Information Criterion function (EBICglasso) was used as an estimator. In the Gaussian graphical model, the partial correlation coefficients were directly used as edge weights between every two nodes in the network; there was no edge if the partial correlation coefficient was 0, e.g., Epskamp et al. [51]. The thickness of edges represents the strength of the relationships between variables; the absence of a line implies no (or a very weak) relationship between the variables. Due to the presence of positive and negative edges, the expected influence centrality index was estimated for all variables. This index provides specific information regarding the impact of each node on the rest of the network [52]: "aim to assess the nature and strength of a node's cumulative influence within the network, and thus the role it may be expected to play in the activation, persistence and remission of the network" (p. 748). In addition, the closeness (the more a node is positioned to the center of the network, the closer it is to all other nodes) and betweenness (number of shortest paths passing through a node or the power of a node to interrupt the flow of information on the network) centrality indices were

calculated. All indices are presented in a standardized way. JASP 0.14 software was used to carry out the analysis for network estimation and visualization, with the auto correlation method (automatically detects the type of input variable and uses the most suitable type of correlation).

## Preliminary analysis

Reliability estimations are presented in Table 1, descriptive statistics in Table 2 and partial correlations in Table 3. The estimation of the reliability of the scores on each scale was adequate to good.

## Network analysis

The centrality indices are shown in Table 4. Positive emotions appear as the most influential variable (with the highest values of positive expected influence indices; Hypothesis 2) in the positive direction of the network for Spanish adolescents who evaluated Romanian youth (1.65), Spanish adolescents who evaluated Ecuadorian youth (2.06), Moroccan (1.82), Romanian (1.37) and Ecuadorian (1.80) adolescents. This variable (positive emotions) was the second most influential for Spanish adolescents who evaluated Moroccan youth (1.12) behind sociability (1.74). The sociability dimension was the second most influential for Romanian (1.23) and Ecuadorian (1.13) adolescents, as was the quantity of contact for Spanish adolescents who evaluated Romanian youth (1.46).

In the negative direction, the immorality dimension appeared to be more influential (with the highest values of negative expected influence indices; Hypothesis 1) for Spanish adolescents who evaluated Moroccan youth (-1.55), Spanish adolescents who evaluated Ecuadorian youth (-1.90), and Moroccan (-1.51) and Romanian (-1.80) adolescents, followed by behavioral tendencies of harm for Moroccan (-1.43) and Romanian (-1.10) adolescents. Harm was the most influential variable for Spanish adolescents who evaluated Romanian youth (-1.32) and Ecuadorian adolescents (-1.33), followed by immorality for both groups (-1.22 and -1.27, respectively).

Regarding closeness indices, positive emotions appeared as the best-connected node within the network (with the highest values of closeness indices; Hypothesis 2) for Spanish

**Table 2. Descriptive statistics of all the variables in the six sub-samples.**

|  | SM group | SR group | SE group | M group | R group | E group |
| --- | --- | --- | --- | --- | --- | --- |
| Variables | *M* (*SD*) | M (*SD*) | M (*SD*) | M (*SD*) | M (*SD*) | M (*SD*) |
| Morality | 2.76 (0.79) | 3.42 (0.74) | 3.45 (0.73) | 3.29 (0.70) | 3.38 (0.58) | 3.06 (0.59) |
| Immorality | 2.98 (0.79) | 2.59 (0.82) | 2.29 (0.76) | 2.60 (0.82) | 2.64 (0.78) | 2.75 (0.75) |
| Sociability | 3.04 (0.85) | 3.64 (0.76) | 3.70 (0.77) | 3.66 (0.74) | 3.94 (0.58) | 3.70 (0.72) |
| Competence | 3.26 (0.74) | 3.62 (0.67) | 3.45 (0.72) | 3.50 (0.65) | 3.53 (0.58) | 3.56 (0.57) |
| Positive emotions | 2.85 (0.85) | 3.45 (0.80) | 3.32 (0.85) | 3.45 (0.70) | 3.57 (0.66) | 3.30 (0.70) |
| Negative emotions | 2.41 (0.86) | 2.04 (0.78) | 1.87 (0.66) | 2.17 (0.65) | 2.10 (0.68) | 2.16 (0.72) |
| Facilitation | 3.30 (0.92) | 3.91 (0.81) | 3.77 (0.87) | 3.94 (0.78) | 4.11 (0.66) | 3.77 (0.73) |
| Harm | 1.51 (0.63) | 1.34 (0.57) | 1.30 (0.43) | 1.56 (0.78) | 1.44 (0.53) | 1.35 (0.41) |
| Quantity of contact | 3.59 (1.09) | 3.91 (0.90) | 3.67 (0.96) | 4.15 (0.99) | 4.50 (0.89) | 3.77 (1.03) |
| Quality of contact | 3.50 (0.76) | 3.92 (0.72) | 3.85 (0.77) | 3.85 (0.76) | 4.10 (0.72) | 3.84 (0.77) |

SM group = Spanish adolescents who evaluate Moroccan-origin youth; SR group = Spanish adolescents who evaluate Romanian-origin youth; SE group = Spanish adolescents who evaluate Ecuadorian-origin youth; M group = Moroccan-origin adolescents; R group = Romanian-origin adolescents; E group = Ecuadorian-origin adolescents.

**Table 3. Partial correlations between all the variables in the six sub-samples.**

|  | SM group | SR group | SE group | M group | R group | E group |
|---|---|---|---|---|---|---|
| M-I | -.25*** | -.32*** | -.29*** | -.27*** | -.32*** | -.34*** |
| M-S | .47*** | .33*** | .38*** | .20** | .36*** | .29*** |
| M-C | .18*** | .23*** | .25*** | .20*** | .32*** | .10 |
| M-PE | .09* | .05 | .16** | .12* | -.07 | .18* |
| M-NE | -.09 | .06 | -.07 | -.07 | -.03 | -.00 |
| M-F | .10* | .08 | .02 | .05 | .04 | -.01 |
| M-H | .07 | .05 | .14* | .13* | .14 | .12 |
| M-QC$_1$ | -.03 | -.03 | .01 | -.06 | .01 | -.01 |
| M-QC$_2$ | -.06 | .14* | -.02 | .13* | .03 | .08 |
| I-S | -.10* | -.18** | -.06 | -.17** | -.05 | .05 |
| I-C | -.00 | -.03 | .08 | -.00 | -.06 | .15 |
| I-PE | -.04 | -.12 | -.09 | .08 | -.14 | -.03 |
| I-NE | .25*** | .32*** | .28*** | .31*** | .41*** | .38*** |
| I-F | .13** | .02 | .04 | .04 | .03 | .04 |
| I-H | .12* | .06 | .04 | .09 | .02 | .06 |
| I- QC$_1$ | .03 | .26*** | .11 | .00 | -.10 | .16* |
| I- QC$_2$ | -.04 | -.01 | -.03 | .04 | .09 | -.10 |
| S-C | .17*** | .26*** | .23*** | .23*** | .21* | .06 |
| S-PE | .25*** | .25*** | .12* | .26*** | .28** | .13 |
| S-NE | .05 | -.01 | -.16** | .06 | -.04 | -.17* |
| S-F | .05 | .06 | .00 | .03 | .01 | .29*** |
| S-H | .06 | -.06 | -.04 | .01 | -.04 | .16* |
| S- QC$_1$ | .09 | .07 | .02 | .12* | .03 | .01 |
| S-QC$_2$ | .18*** | -.02 | .11 | .06 | .08 | .16* |
| C-PE | .07 | .03 | .12* | .21*** | .09 | .17* |
| C-NE | .06 | .05 | .11 | -.06 | .15 | .02 |
| C-F | .10* | -.00 | -.02 | .01 | -.03 | .15* |
| C-H | -.14** | -.10 | -.11 | -.01 | -.18 | .04 |
| C-QC$_1$ | -.08 | .08 | .04 | -.06 | -.09 | .06 |
| C-QC$_2$ | -.01 | .00 | .02 | -.07 | .07 | .01 |
| PE-NE | -.14** | -.09 | .07 | -.12* | .14 | .06 |
| PE-F | .41*** | .48*** | .55*** | .33*** | .50*** | .33*** |
| PE-H | .09 | .23*** | -.08 | .01 | .13 | -.04 |
| PE-QC$_1$ | .03 | .05 | .00 | .04 | -.05 | .18 |
| PE-QC$_2$ | .20*** | .35*** | .28*** | .21*** | .25** | .12 |
| NE-F | -.10* | -.04 | -.10 | .11 | .07 | .06 |
| NE-H | .35*** | .32*** | .45*** | .31*** | .37*** | .36*** |
| NE-QC$_1$ | .14** | .20** | -.00 | -.05 | -.11 | .11 |
| NE-QC$_2$ | -.07 | -.26*** | -.15* | -.03 | -.28** | -.05 |
| F-H | -.24*** | -.45*** | -.09 | -.19** | -.36*** | -.35*** |
| F-QC$_1$ | .24*** | .16** | .23*** | .21*** | .08 | .03 |
| F-QC$_2$ | .06 | -.08 | .06 | .22*** | .09 | .07 |
| H-QC$_1$ | .11* | -.01 | .10 | -.06 | .07 | .02 |
| H-QC$_2$ | -.18*** | -.02 | .03 | -.11 | .02 | .04 |

(*Continued*)

**Table 3.** (Continued)

|  | SM group | SR group | SE group | M group | R group | E group |
|---|---|---|---|---|---|---|
| $QC_1$- $QC_2$ | .27*** | .33*** | .26*** | .22*** | .44*** | .30*** |

*$p < .05$;

**$p < .01$;

***$p < .001$;

SM group = Spanish adolescents who evaluate Moroccan-origin youth; SR group = Spanish adolescents who evaluate Romanian-origin youth; SE group = Spanish adolescents who evaluate Ecuadorian-origin youth; M group = Moroccan-origin adolescents; R group = Romanian-origin adolescents; E group = Ecuadorian-origin adolescents; M = Morality; I = Immorality; S = Sociability; C = Competence; PE = Positive emotions; NE = Negative emotions; F = Facilitation; H = Harm; $QC_1$ = Quantity of contact; $QC_2$ = Quality of contact.

adolescents who evaluated Moroccan youth (1.42), Spanish adolescents who evaluated Romanian youth (1.30), and Moroccan (1.71), Romanian (1.50) and Ecuadorian (1.45) adolescents. The positive emotions node was also the second best connected variable (0.99) behind morality (1.47) for Spanish adolescents who evaluated Ecuadorian youth. Scores in this index were followed by behavioral tendencies of facilitation for Spanish adolescents who evaluated Moroccan youth (1.20), and Moroccan adolescents (1.36), and by sociability for Romanian (1.10) and Ecuadorian (1.15) adolescents.

Finally, positive emotions appeared as the node with the highest scores in betweenness (there were more short paths that passed through this node than through the other ones; Hypothesis 2) for Spanish adolescents who evaluated Romanian and Ecuadorian youths (1.72 and 1.75, respectively) and for Romanian and Ecuadorian adolescents (1.67 and 1.44, respectively). This node was followed by sociability and immorality for Spanish adolescents who evaluated Romanian youth (1.03), by sociability for Romanian adolescents (0.91), by negative emotions for Spanish adolescents who evaluated Ecuadorian youth (1.43) and by behavioral tendencies of facilitation for Ecuadorian adolescents (1.13). For Spanish adolescents who

**Table 4. Centrality indices of each variable in the six sub-samples.**

|  | SM group | | | SR group | | | SE group | | | M group | | | R group | | | E group | | |
|---|---|---|---|---|---|---|---|---|---|---|---|---|---|---|---|---|---|---|
|  | EIn | Clos | Bet | EIn | Clos | Bet | EIn | Clos | Bet | EIn | Clos | Bet | EIn | Clos | Bet | EIn | Clos | Bet |
| M | -0.24 | 0.44 | 0.03 | -0.14 | -0.08 | -0.34 | -0.20 | **1.47** | 0.16 | -0.42 | 0.40 | 0.31 | -0.70 | 0.46 | 0.15 | -0.80 | 0.67 | 0.83 |
| I | **-1.55** | -0.76 | -0.32 | **-1.22** | 0.55 | **1.03** | **-1.90** | 0.01 | -0.48 | **-1.51** | -0.64 | -0.10 | **-1.80** | 0.11 | -0.10 | **-1.27** | -0.37 | -0.09 |
| S | **1.74** | **1.07** | **1.09** | 0.59 | 0.49 | **1.03** | 0.37 | 0.86 | -0.48 | 0.74 | 0.60 | -0.92 | **1.23** | **1.10** | **0.91** | **1.13** | **1.15** | 0.52 |
| C | -0.51 | -1.67 | -1.37 | -0.03 | -2.34 | -1.38 | 0.15 | -0.81 | -1.12 | 0.09 | -0.24 | -0.92 | 0.18 | -1.03 | -1.37 | -0.36 | -1.35 | -1.32 |
| PE | **1.12** | **1.42** | 0.39 | **1.65** | **1.30** | **1.72** | **2.06** | 0.99 | **1.75** | **1.82** | **1.71** | **1.54** | **1.37** | **1.50** | **1.67** | **1.80** | **1.45** | **1.44** |
| NE | -0.36 | -0.39 | 0.39 | -0.20 | 0.75 | 0.34 | -0.54 | -0.12 | **1.43** | -0.13 | -0.63 | -0.10 | 0.01 | 0.42 | 0.66 | 0.15 | -0.72 | -0.40 |
| F | 0.40 | **1.20** | **1.44** | -0.71 | -0.07 | -0.34 | 0.41 | 0.29 | 0.16 | 0.10 | **1.36** | **1.74** | -0.16 | 0.39 | -0.61 | 0.20 | **1.07** | **1.13** |
| H | -0.98 | -0.18 | 0.74 | **-1.32** | -0.59 | -0.69 | -0.63 | -1.04 | -0.80 | **-1.43** | -0.36 | 0.31 | **-1.10** | -0.71 | -0.61 | **-1.33** | -0.25 | 0.21 |
| $QC_1$ | 0.77 | -0.61 | -1.02 | **1.46** | -0.44 | -1.03 | 0.01 | -1.81 | -1.12 | -0.04 | -1.49 | -0.92 | 0.21 | -1.80 | -1.37 | -0.02 | -0.97 | -1.32 |
| $QC_2$ | -0.37 | -0.52 | -1.37 | -0.09 | 0.42 | -0.34 | 0.27 | 0.16 | 0.48 | 0.77 | -0.71 | -0.92 | 0.74 | -0.43 | 0.66 | 0.50 | -0.69 | -1.01 |

SM group = Spanish adolescents who evaluate Moroccan-origin youth; SR group = Spanish adolescents who evaluate Romanian-origin youth; SE group = Spanish adolescents who evaluate Ecuadorian-origin youth; M group = Moroccan-origin adolescents; R group = Romanian-origin adolescents; E group = Ecuadorian-origin adolescents; EIn = Expected influence; Clos = Closeness; Bet = Betweenness; M = Morality; I = Immorality; S = Sociability; C = Competence; PE = Positive emotions; NE = Negative emotions; F = Facilitation; H = Harm; $QC_1$ = Quantity of contact; $QC_2$ = Quality of contact; Numbers in bold indicate the highest scores for the different centrality indices.

evaluated Moroccan youth and for Moroccan-origin adolescents, facilitation behavior was the node with the higher scores in this index (1.44 and 1.74, respectively), followed by sociability for Spanish adolescents who evaluated Moroccan youth (1.09) and by positive emotions for Moroccan adolescents (1.54). The resulting network is displayed in Fig 1 (for Spanish adolescents) and Fig 2 (for adolescents with immigrant backgrounds). In general, for the six subsamples, we found a similar pattern in the structure of the network. The nodes corresponding to the stereotypes appear as a structure at the top of the networks in all samples. Similarly, nodes corresponding to positive and negative emotions appear below the stereotype structure. At the bottom of the networks, facilitation and harm nodes appear at both ends and are related to emotions. Finally, the contact quality node appears, also in the lower part of the network, and, in general, related to the facilitation and positive emotions nodes.

Morality, sociability and competence dimensions were directly and positively related to each other for Spanish adolescents who evaluated Moroccan, Romanian and Ecuadorian youths, and Moroccan and Romanian adolescents. For Ecuadorian adolescents, only morality and sociability were related. Sociability is the stereotype dimension that had the strongest direct relationship with positive emotions for Spanish adolescents who evaluated Moroccan and Romanian youths, and Moroccan and Romanian adolescents. For Spanish adolescents who evaluated Ecuadorian youth, morality was more strongly related to positive emotions than sociability. For Ecuadorian adolescents, only morality and competence were related to positive emotions. As can be seen on the network (Hypothesis 3), emotions act as a link between stereotype dimensions (at the top of the network) and behavioral tendencies (at the bottom and sides of the network): positive emotions act as a link between the positive stereotypical dimensions (sociability, morality and competence) and behavioral tendencies of facilitation whereas negative emotions act as a link between the immorality dimension and behavioral tendencies of harm.

Quantity and quality of contact were directly and positively related to each other in the six subsamples. Quality of contact was related to the rest of the network (Hypothesis 4), mainly through positive emotions for the Spanish adolescents who evaluated Moroccan and Ecuadorian youths, and Moroccan, Romanian and Ecuadorian adolescents. For the Spanish adolescents who evaluated Romanian youth, the quality of contact was related to the rest of the network through positive and negative emotions.

## Discussion

The aim of this study was to analyze the relationship between different evaluative reactions of the intergroup attitudes (stereotypes, emotions, behavioral tendencies) and contact in Spanish adolescents evaluating different ethnic minorities (Moroccans, Romanians and Ecuadorians) and in immigrant-background adolescents evaluating Spanish youth, using empirical network analysis to explore the structure and dynamic properties of intergroup attitudes in these groups.

The centrality indices showed that immorality appears as the most influential stereotype dimension, following by sociability (above morality and competence). These results partially confirm our first hypothesis (H1). The result of the prominent role of immorality above the other dimensions (morality, sociability and competence) is innovative but not surprising [13–17]. Many previous studies have also shown a prominent role of morality in intergroup relations compared to other dimensions such as sociability and competence [10, 11, 53]. In our study, however, sociability had a more central role than morality. A possible explanation for the differences found with respect to previous studies could be the simultaneous measurement of both variables (morality and immorality) and the control of the effect of the rest of the

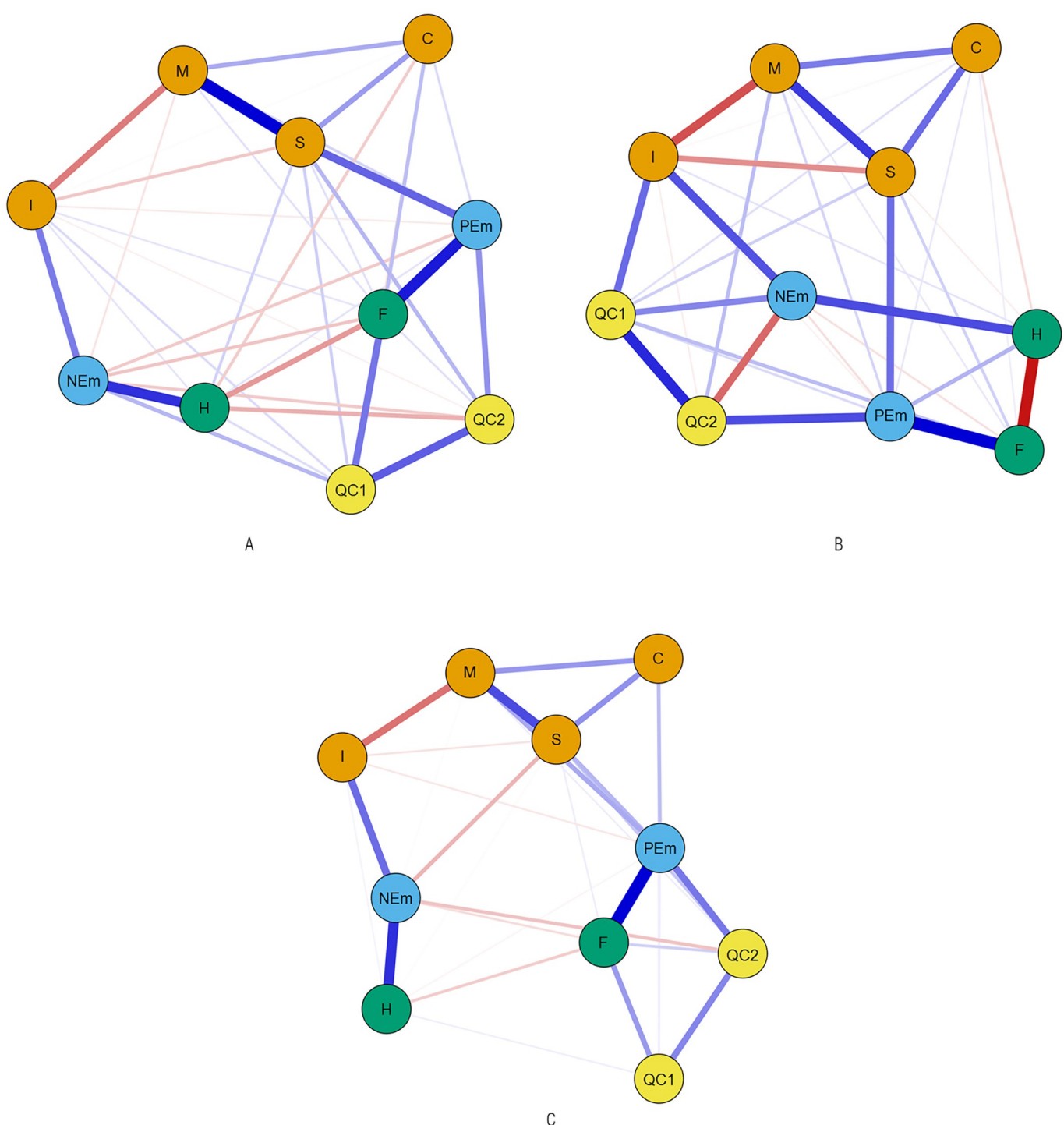

**Fig 1.** Estimated networks for Spanish adolescents who evaluate Moroccan (A), Romanian (B) and Ecuadorian (C) youths. M = Morality; I = Immorality; S = Sociability; C = Competence; PE = Positive emotions; NE = Negative emotions; F = Facilitation; H = Harm; QC$_1$ = Quantity of contact; QC$_2$ = Quality of contact.

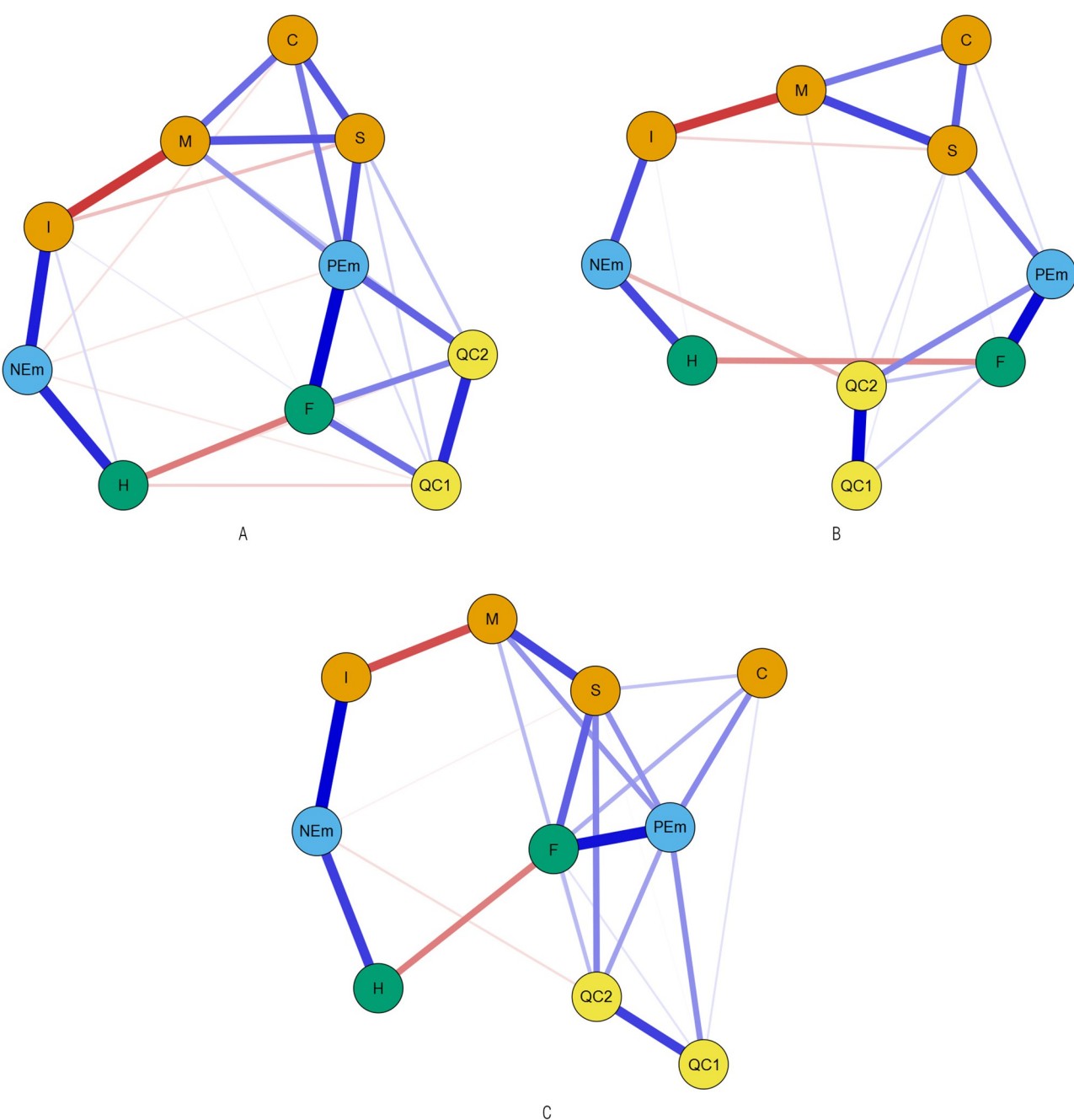

**Fig 2.** Estimated networks for Moroccan (A), Romanian (B) and Ecuadorian (C) adolescents who evaluate Spanish youth. M = Morality; I = Immorality; S = Sociability; C = Competence; PE = Positive emotions; NE = Negative emotions; F = Facilitation; H = Harm; $QC_1$ = Quantity of contact; $QC_2$ = Quality of contact.

evaluative reactions of attitudes due to the use of partial correlations in network analysis. In addition, most previous studies based on the SCM and its subsequent modifications have focused on the adult population [10]. In a study by Constantin and Cuadrado [23], which was conducted on an adolescent population, the results showed that sociability was moderately more important than morality in predicting positive emotions. It is possible that, for

adolescents, sociability plays a more prominent role than morality, contrary to what happens with the adult population. Despite this, the present study is in line with previous findings on the distinction between sociability and morality and covered in the Moral Primacy Model [25].

In general, the results show similar structural patterns in the six studied groups: the positive emotions node was the most highly connected node (or central) with the rest of the network, and two types of emotions (positive and negative) acted as links between stereotypes and behavioral tendencies. These results are in line with the BIAS Map [5, 6], the IET [19] and research derived from these models [22] and confirming our second hypothesis (H2). The central role of emotions has also been found by Nariman et al. [30] from a network approach. In their study, they focused on stereotypical, emotional and behavioral evaluative responses toward Roma people (one of the main minority groups in Spain and in many European countries, associated with the stereotype of begging and delinquency, and characterized as a target group for discrimination, hate crimes and social exclusion [54–56]) and they observed that the central nodes are those of an affective nature (national identity, sympathy, and empathy).

In the present study, using network analysis, we found a relationship between stereotypes and behavioral tendencies through emotions, but this was compatible with the bidirectionality found in previous research [22] and defended by previous theories (IET [19]). They also did not find a direct relationship between sociability and positive emotions. In our study, on the contrary, the dimension of sociability was the closest and the one that presented the strongest direct relationship with positive emotions for the four groups in the study: Spanish adolescents evaluating Moroccan and Romanian youths, and Moroccan and Romanian origin adolescents evaluating Spanish youth. Even so, the direct relationship between morality and positive emotions did not appear in two of the studied groups (Spaniards evaluating Romanian youth and Romanian-origin adolescents) or was very week (for the rest of the groups).

Although morality and immorality dimensions are directly related, they also appear as dimensions that are part of different processes: while sociability (primarily) and morality and competence (to a lesser extent) are related to facilitation behaviors through positive emotions, immorality is related to harm behaviors through negative emotions, confirming our third hypothesis (H3). These results are in line with previous studies [26, 27], which pointed out the need to study positive and negative attitudes as separate dimensions. Including the stereotype dimension of immorality and analyzing its relationship with behaviors of harm through negative emotions is an important contribution of this study. Positive emotions also show a key role in the relationship between contact and intergroup attitudes. The results show some common patterns in the different groups. For the Spaniards evaluating Moroccan and Ecuadorian youths, and for the three groups of adolescents with an immigrant background (Moroccan, Romanian and Ecuadorian), the quality of contact was related to the rest of the network, mainly through positive emotions. For Spanish adolescents evaluating Romanian youth, this variable was related to the rest of the network through positive and negative emotions. These results partially confirm our hypothesis (H4) and they are in line with previous research [37, 38] showing that the quality of contact is related with different measures of intergroup attitude, especially affective reactions. The quality of contact seems to have a greater influence than the quantity (H4) for Spanish adolescents evaluating Ecuadorian youth and for the three groups of adolescents with an immigrant background. This result coincides with previous findings [37–39], showing that the quality of contact has a greater effect on intergroup attitudes than quantity. However, for the Spaniards who evaluated Moroccan youth, the quantity of contact had a greater centrality in the network than quality.

Taking into account the structure and dynamic properties of the networks and the estimated centrality indices, some conclusions can be drawn that may be of interest when

planning interventions. Firstly, stereotypes relate to behavioral tendencies through two different pathways: through positive emotions for facilitation behaviors and through negative emotions for harm behaviors. To achieve successful intergroup relations involving cooperation and the development of friendly relationships (and not just a reduction of conflict), it would be appropriate to intervene in parallel in these two pathways. Secondly, due to the centrality of these variables and, therefore, their ability to affect the entire network, interventions should focus on positive emotions (and the perception of sociability) towards out-groups and decrease the perception of immorality. This could be an appropriate strategy to achieve positive overall attitudes and intergroup relationships with more and better contact. In this study, it is observed that intergroup contact quality plays a key role in improving intergroup attitudes. The direct relationship between the quality of contact and positive emotions (one of the most central variables of the network) indicates that achieving quality relationships could have an important effect on the whole attitudinal network. For future research, it would be of great interest to inquire about the people with whom adolescents have contact in class (whether they are from their own ethnocultural group or from other out-groups), including a social network analysis perspective [57, 58].

Surveying adolescents outside of the school context is very difficult, so the use of accessible samples (instead of a representative sample) and the sample size of some groups requires caution regarding the generalization of results to the rest of the adolescent population, which may be a limitation of the present study. In addition, it should be noted that all the measures used are self-reported, capturing only the self-perceptions of adolescents, with the drawbacks that this type of measure can have in capturing intergroup attitudes already described in the literature.

Despite these limitations, the present study contributes significantly to the psychosocial literature, using empirical network analysis, a recent and innovative perspective in the analysis of the structure of intergroup attitudes that allowed us to identify the key variables of this structure. Furthermore, this analysis was carried out with six groups of adolescents, from the majority (Spanish) and minority perspective (Moroccan, Romanian, and Ecuadorian origin). Our study shows similarities in the structure of intergroup attitudes and their key variables (e.g., the central role of emotions connecting stereotypes and behavioral tendencies), but also some differences depending on the group, thus supporting the specificity of intergroup attitudes depending on the context and related groups.

## Author Contributions

**Conceptualization:** María Sánchez-Castelló, Marisol Navas, Antonio J. Rojas.

**Data curation:** María Sánchez-Castelló, Antonio J. Rojas.

**Formal analysis:** María Sánchez-Castelló, Antonio J. Rojas.

**Funding acquisition:** Marisol Navas, Antonio J. Rojas.

**Investigation:** María Sánchez-Castelló, Marisol Navas, Antonio J. Rojas.

**Methodology:** María Sánchez-Castelló, Marisol Navas, Antonio J. Rojas.

**Project administration:** Marisol Navas, Antonio J. Rojas.

**Resources:** Marisol Navas, Antonio J. Rojas.

**Supervision:** Marisol Navas, Antonio J. Rojas.

**Writing – original draft:** María Sánchez-Castelló, Marisol Navas, Antonio J. Rojas.

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
