## [Decision Letter · Decision Letter 0]

15 Mar 2022

PONE-D-21-14451Intergroup attitudes and contact between Spanish and immigrant-background adolescents using network analysis.PLOS ONE

Dear Dr. Sánchez Castelló,

Thank you for submitting your manuscript to PLOS ONE. After careful consideration, we feel that it has merit but does not fully meet PLOS ONE’s publication criteria as it currently stands. Therefore, we invite you to submit a revised version of the manuscript that addresses the points raised during the review process.

The reviewers raised a number of concerns regarding the study design, the description of variables, the lack of a control, and the lack of a clear hypothesis in the main text of the manuscript. Their comments can be viewed in full, below and in the attached file.

We look forward to receiving your revised manuscript.

Kind regards,

Natasha McDonald, PhD

Associate Editor

PLOS ONE

“This study is part of the project ‘Prejudiced attitudes, acculturation process and adjustment of immigrant and host adolescents’ [Reference PS2016-80123-P], funded by the Ministry of Economy, Industry and Competitiveness (Spain).”

“This study is part of the project ‘Prejudiced attitudes, acculturation process and adjustment of immigrant and host adolescents’ [Reference PS2016-80123-P], funded by the Ministry of Economy, Industry and Competitiveness (Spain).

Reviewers' comments:

Reviewer's Responses to Questions

**Comments to the Author**

1. Is the manuscript technically sound, and do the data support the conclusions?

Reviewer #1: Partly

Reviewer #2: Yes

Reviewer #3: Yes

2. Has the statistical analysis been performed appropriately and rigorously? 

Reviewer #1: I Don't Know

Reviewer #2: Yes

Reviewer #3: Yes

3. Have the authors made all data underlying the findings in their manuscript fully available?

Reviewer #1: Yes

Reviewer #2: Yes

Reviewer #3: Yes

4. Is the manuscript presented in an intelligible fashion and written in standard English?

Reviewer #1: Yes

Reviewer #2: Yes

Reviewer #3: Yes

5. Review Comments to the Author

Reviewer #1: The authors present an interesting article that draws on a large sample of Spanish adolescents (an understudied population given the topic) with necessary considerations taken regarding ethics and the surveying of minors. I truly enjoyed reading the article.

Here are some considerations that might help make it even better:

CONTEXT AND PARTICIPANTS

Some more on the context might be relevant. For non-Spanish readers “about 10% of the total Student body during the 2019-2020...” (p. 3) fails to highlight the 2000-2020 increase in the immigrant population at large and in schools within Spain. What are the numbers like in the specific area in Spain where data was collected? Relatedly, where were these public secondary schools located? In larger or smaller cities? Within what Autonomous Community or Autonomous Communities? Were all schools from the same city or county? Were they diverse in their percentage of immigrant population? How many total schools are we talking about? How is this area different or similar to other in Spain. Were the Spanish, Moroccan, Romanian and Ecuadorian groups surveyed similar the population of study? If so, could this be highlighted in a table for example? Also, certain language could use some clarification. For example, I understand what the authors mean by “Roma people” but some context for non-European readers might be helpful.

LITERATURE REVIEW

Overall the citations seem a bit dated and few studies in Spain are cited. There is some high-quality recent (2018-2021) research coming out of nearby Catalonia (Spain) that I think the authors might want to read and integrate. These studies also cite other relevant research you might want to look into, for example:

Bobowik, M., Benet-Martínez, V., & Repke, L. (2021). “United in diversity”: The interplay of social network characteristics and personality in predicting outgroup attitudes. Group Processes & Intergroup Relations, https://doi.org/10.1177/13684302211002918

Bobowik, M., Benet‐Martínez, V., & Repke, L. (2021, online first). Ethnocultural diversity of immigrants' personal social networks, bicultural identity integration and global identification. International Journal of Psychology. https://doi.org/10.1002/ijop.12814

Repke, L., & Benet-Martínez, V. (2018). The (diverse) company you keep: Content and structure of immigrants’ social networks as a window into intercultural relations in Catalonia. Journal of Cross-Cultural Psychology, 49(6), 924-944.

Wilson-Daily, A. E., & Kemmelmeier, M. (2021). Who is on our side? complexities of national identification among native and immigrant youth in Catalonia. Journal of Youth Studies, 24(8), 994-1014.

Wilson-Daily, A. E., Kemmelmeier, M., & Prats, J. (2018). Intergroup contact versus conflict in Catalan high schools: A multilevel analysis of adolescent attitudes toward immigration and diversity. International Journal of Intercultural Relations, 64, 12-28.

Also, a bit dated is the reference to Allport. Athough Pettigrew and Tropp (2006) is mentioned, their building off of and improvement of Allport’s original work is not referenced. Pettigrew and Tropp expanded Allport’s original hypothesis and highlighted necessities for its accomplishment. Nevertheless, this is not highlighted in any way in the article. Furthermore, the authors might be interested in consulting and considering some research from Hewstone and colleagues, such as:

Laurence, J., Schmid, K., Rae, J. R., Hewstone, M. (2019). Prejudice, contact, and threat at the diversity–segregation nexus: A cross-sectional and longitudinal analysis of how ethnic out-group size and segregation interrelate for inter-group relations. Social Forces, 97, 1029–1066. https://doi.org/10.1093/sf/soy079

Wölfer, R., Faber, N. S., Hewstone, M. (2015). Social network analysis in the science of groups: Cross-sectional and longitudinal applications for studying intra- and intergroup behavior. Group Dynamics: Theory, Research, and Practice, 19, 45–61. https://doi.org/10.1037/gdn0000021

Wölfer, R., Hewstone, M. (2017). Beyond the dyadic perspective: 10 reasons for using social network analysis in intergroup contact research. British Journal of Social Psychology, 56, 609–617. https://doi.org/10.1111/bjso.12195

These should be at least read (if not cited) and perhaps contemplated in regard to study limitations. I see that you do reference a 1993 article of Hewstone’s but I would encourage a more in-depth reading of his group’s more recent work.

ANALYSES

I do not see any control for example for socioeconomic status, given the importance that this plays on intergroup attitudes, the autors might want to comment on this. If this data is available I would find a way to include it in the analyses.

Also, if students are nested in classrooms, which are then nested in schools, and then neighborhoods, a study on intergroup contact might want to contemplate multilevel modeling. Was this data collected (i.e., who is in what classroom/School)? The analysis might benefit from “who am I in contact with in this classroom /school” in ADDITION to the self-reported “Quality of contact”.

OTHER, MORE MINOR QUESTIONS OR CONSIDERATIONS

If students did not identify as Spanish, Moroccan, Romanian or Ecuadorian did they not fill out a questionnaire? How were the diferent survey options presented to the students?

Reviewer #2: The manuscript investigates the perception of different ethnic minorities in light of the SCM and Bias Map. The manuscript is well written and easy to follow. The study considers un understudied sample (adolescents) and employed an innovative approach (network analysis). Therefore, I am happy to recommend this paper for publication after a revision.

1) The distinction between sociability and morality has been dismissed lightly. I would suggest the authors better describe the innovative value of their theoretical approach. They could describe the moral primacy model of impression development (Brambilla et al., 2021; Goodwin, 2015) and state how the present study fits/complements/extends prior insights on the distinction between sociability and morality.

Brambilla, M., Sacchi, S., Rusconi, P., Goodwin, G. (2021). The primacy of morality in impression development: Theory, research, and future directions. Advances in Experimental Social Psychology, 64, 187-262.

Goodwin, G. P. (2015). Moral character in person perception. Current Directions in Psychological Science, 24, 38-44.

2) Hypotheses: The distinction between morality and immorality is well explained in the general discussion. However, I would suggest describing such a distinction only in the introduction. This would be functional to better understand the hypotheses. I would refer to prior work on the confirmability of traits-concepts (Reeder & Brewer, 1979).

Reeder, G. D., & Brewer, M. B. (1979). A schematic model of dispositional attribution in interpersonal perception. Psychological Review, 86, 61.

3) I do not immediately understand why the authors measured the negative and the positive pole of morality, while sociability and competence were measured by considering only the positive pole. Please explain.

4) I would also suggest describing more in detail how morality has been conceptualized.

Morality is a broad construct defined by distinct components (see for instance the distinction between deontology and consequentialism - Sacchi et al., 2014; or the distinction between harm and fairness, Gray et al., 2009).

Sacchi, S., Riva, P., Brambilla, M., & Grasso, M. (2014). Moral reasoning and climate change mitigation: The deontological reaction toward the market-based approach. Journal of Environmental Psychology, 38, 252-261.

Gray, K., & Wegner, D. M. (2009). Moral typecasting: divergent perceptions of moral agents and moral patients. Journal of personality and social psychology, 96(3), 505.

Reviewer #3: This is a very interesting paper that contributes to the intergroup attitudes literature. It is well-performed and results are clearly exposed.

As for suggestions that could improve the paper are: including in the abstract the application of the results, deepen on the conclusions as for the implications of the results, and performing further analyses to guarantee the comparability of first and secon-generation of the group members. Moreover, were there any difference by different schools? maybe the incomes, sex or tother variables should also be considered.

6. PLOS authors have the option to publish the peer review history of their article (what does this mean?). If published, this will include your full peer review and any attached files.

Reviewer #1: No

Reviewer #2: No

Reviewer #3: **Yes: **Esther Lopez-Zafra

---

## [Author Response · Author response to Decision Letter 0]

28 Apr 2022

Dear Editor and Reviewers,

Thank you very much for the consideration and the feedback you have provided to our manuscript. We have introduced your recommendations in this new version and we explain how we addressed each advice or question in the following pages.

REVIEWER #1: 

The authors present an interesting article that draws on a large sample of Spanish adolescents (an understudied population given the topic) with necessary considerations taken regarding ethics and the surveying of minors. I truly enjoyed reading the article. Here are some considerations that might help make it even better: 

Context and participants

Some more on the context might be relevant. For non-Spanish readers “about 10% of the total Student body during the 2019-2020...” (p. 3) fails to highlight the 2000-2020 increase in the immigrant population at large and in schools within Spain. What are the numbers like in the specific area in Spain where data was collected? Relatedly, where were these public secondary schools located? In larger or smaller cities? Within what Autonomous Community or Autonomous Communities? Were all schools from the same city or county? Were they diverse in their percentage of immigrant population? How many total schools are we talking about? How is this area different or similar to other in Spain. Were the Spanish, Moroccan, Romanian and Ecuadorian groups surveyed similar the population of study? If so, could this be highlighted in a table for example?

We have added some information in the manuscript as detailed in the following paragraphs about a more in-depth description of the context in which the study was conducted. The questionnaires were applied in 17 secondary schools located in five provinces in Spain (Almería, Alicante, Castellón, Madrid and Murcia), selected because they are among those with the highest percentages of foreign population from any of the three origins (Moroccan, Romanian or Ecuadorian). Specifically, the percentages of foreign population in these provinces were 21.37%, 19.84%, 15.18%, 14.15% and 14.64%, respectively (National Institute of Statistics, [47]).

All the municipalities in which the questionnaires were applied has less than 100000 inhabitants, except Puente de Vallecas (Madrid) with 240000 inhabitants. Official data on the percentages of foreign adolescents enrolled in all secondary schools was not available. 

 Information included in the manuscript:

Introduction section (lines 26-30): “The increase in diversity in current societies is reflected in the presence of students with an immigrant background in secondary schools (in Spain, about 10% of the total student body during the 2019-2020 academic year whereas this percentage was less than 2% in the 2000-2001 academic year, according to the Spanish Ministry of Education, Culture and Sport [1, 2]).”

Participants section (lines 201-202): “They were enrolled in 17 public secondary schools in different municipalities in five Spanish provinces (Almería, Alicante, Castellón, Madrid and Murcia).”

Procedure section (lines 265-269): “After carrying out a residential census analysis, five sampling zones (Almería, Alicante, Castellón, Madrid and Murcia) were chosen with the largest immigrant population [34]. Specifically, the percentages of foreign population in these provinces were 21.37%, 19.84%, 15.18%, 14.15% and 14.64%, respectively according to the National Institute of Statistics [48].”

Also, certain language could use some clarification. For example, I understand what the authors mean by “Roma people” but some context for non-European readers might be helpful.

A clarification has been included in parentheses (lines 412-414): “…Roma people (one of the main minority groups in Spain and in many European countries, associated with the stereotype of begging and delinquency, and characterized as a target group for discrimination, hate crimes and social exclusion [54-56])…” 

Literature review

Overall the citations seem a bit dated and few studies in Spain are cited. There is some high-quality recent (2018-2021) research coming out of nearby Catalonia (Spain) that I think the authors might want to read and integrate. These studies also cite other relevant research you might want to look into, for example:

Bobowik, M., Benet-Martínez, V., & Repke, L. (2021). “United in diversity”: The interplay of social network characteristics and personality in predicting outgroup attitudes. Group Processes & Intergroup Relations, https://doi.org/10.1177/13684302211002918

Bobowik, M., Benet‐Martínez, V., & Repke, L. (2021, online first). Ethnocultural diversity of immigrants' personal social networks, bicultural identity integration and global identification. International Journal of Psychology. https://doi.org/10.1002/ijop.12814

Repke, L., & Benet-Martínez, V. (2018). The (diverse) company you keep: Content and structure of immigrants’ social networks as a window into intercultural relations in Catalonia. Journal of Cross-Cultural Psychology, 49(6), 924-944.

Wilson-Daily, A. E., & Kemmelmeier, M. (2021). Who is on our side? complexities of national identification among native and immigrant youth in Catalonia. Journal of Youth Studies, 24(8), 994-1014.

Wilson-Daily, A. E., Kemmelmeier, M., & Prats, J. (2018). Intergroup contact versus conflict in Catalan high schools: A multilevel analysis of adolescent attitudes toward immigration and diversity. International Journal of Intercultural Relations, 64, 12-28.

Thank you very much for the recommendation of these interesting papers. We have integrated the results of some of these studies in our manuscript:

Lines 155-164: “Bobowik et al. [40] conducted a study using a social network approach in Spain with adults of immigrant background and found that the proportion of intergroup contacts (with native-born population) representing strong ties (i.e., close relationships such as friends, romantic partners, and immediate and extended family members) and ethnic diversity among these ties were associated with more favorable attitudes toward outgroups. In another study conducted in the same community with 15- and 16-year-old students, it was observed that students who reported frequent interaction with other students of different nationalities and religions held more positive intergroup attitudes and the presence of a greater number of immigrants in classrooms was related to lower levels of xenophobia [41].”

Also, a bit dated is the reference to Allport. Athough Pettigrew and Tropp (2006) is mentioned, their building off of and improvement of Allport’s original work is not referenced. Pettigrew and Tropp expanded Allport’s original hypothesis and highlighted necessities for its accomplishment. Nevertheless, this is not highlighted in any way in the article. Furthermore, the authors might be interested in consulting and considering some research from Hewstone and colleagues, such as:

Laurence, J., Schmid, K., Rae, J. R., Hewstone, M. (2019). Prejudice, contact, and threat at the diversity–segregation nexus: A cross-sectional and longitudinal analysis of how ethnic out-group size and segregation interrelate for inter-group relations. Social Forces, 97, 1029–1066. https://doi.org/10.1093/sf/soy079

Wölfer, R., Faber, N. S., Hewstone, M. (2015). Social network analysis in the science of groups: Cross-sectional and longitudinal applications for studying intra- and intergroup behavior. Group Dynamics: Theory, Research, and Practice, 19, 45–61. https://doi.org/10.1037/gdn0000021

Wölfer, R., Hewstone, M. (2017). Beyond the dyadic perspective: 10 reasons for using social network analysis in intergroup contact research. British Journal of Social Psychology, 56, 609–617. https://doi.org/10.1111/bjso.12195

These should be at least read (if not cited) and perhaps contemplated in regard to study limitations. I see that you do reference a 1993 article of Hewstone’s but I would encourage a more in-depth reading of his group’s more recent work.

Information regarding intergroup contact has been extended in the manuscript (lines 137-143): “equal status between the groups, common goals, intergroup cooperation, and the support of authorities, law, or custom). However, in Pettigrew and Tropp’s meta-analysis, it was observed that these conditions were not necessary for a reduction in prejudice to occur and these authors proposed that the simple familiarity that occurs with contact generates liking. In addition, they point out that the relationship between familiarity and liking could be influenced by variables such as the reduction of uncertainty, intergroup anxiety and threat”. 

Thank you very much for the recommendations of these articles. We have included one of them in the Introduction section (lines 143-147): “The study by Laurence et al. [36] showed that increased out-group size was related to intergroup attitudes towards them only in more segregated communities (but not in the case of more integrated communities), and that this is explained by an increase in perceived threat and a decrease in the quality of contact (when the increase in out-group size exceeds a certain value).” 

The other two papers have been mentioned in future lines of research (lines 464-467): “For future research, it would be of great interest to inquire about the people with whom adolescents have contact in class (whether they are from their own ethnocultural group or from other out-groups), including a social network analysis perspective [57, 58].”

We have not described these articles in greater depth as they focus on social network analysis, which encompasses network analysis from a psycho-sociological point of view (of great interest to social psychology), while the empirical network analysis we have used is approached from a more psychometric focus with variables of great impact in the psychosocial literature. However, due to their interest, we consider it important to consider them as future lines of research.

Analyses

I do not see any control for example for socioeconomic status, given the importance that this plays on intergroup attitudes, the authors might want to comment on this. If this data is available I would find a way to include it in the analyses.

We have a measure of perceived socioeconomic status in the study (the scores of this variable range from 1 to 10). Following the suggestion of the reviewer, we did perform the corresponding analyses to explore if there were differences between the groups in this variable. The average score is 6.38 (SD = 1.36) for SM group, 6.39 (SD = 1.29) for SR group, 6.43 (SD = 1.33) for SE group, 6.14 (SD = 1.61) for M group, 6.36 (SD = 1.28) for R group, and 6.69 (SD = 1.36) for E group. Results showed that differences between groups appeared only between the E group (Ecuadorian adolescents) and the rest of the groups (p < .001 with SM, SR, SE groups and p < .01 with M and R groups). 

Results of the ANOVA (See attached file titled Response to Reviewers)

One of the ways to include a variable as a covariate in network analysis would be to include this variable as another node within the network and see if substantial changes occur in the proposed network. We have included the perceived socioeconomic status variable in our network analysis (See attached file titled Response to Reviewers)

Perceived socioeconomic status appears, in general, unlinked to the rest of the network and does not produce important changes in the structure of attitudes. There is only a stronger relationship between perceived socioeconomic status and quantity of contact in the SR group (the higher the perceived status of Spanish adolescents, the lesser contact they have with adolescents of Romanian origin). However, the network continues to maintain the same structure presented without introducing it.

For these reasons, we have decided not to include this information in the manuscript so as not to extend the length of the manuscript excessively. However, if the reviewer considers it appropriate, we could add it in a footnote. 

In the participants' section of the new version of the article we have included descriptive data (means and standard deviations) on perceived socioeconomic status, where it can be seen that the scores are similar in all the groups studied (lines 208-211): “Average perceived socioeconomic status score (ranging from 1 to 10 points) was 6.38 (SD = 1.36) for SM group, 6.39 (SD = 1.29) for SR group, 6.43 (SD = 1.33) for SE group, 6.14 (SD = 1.61) for M group, 6.36 (SD = 1.28) for R group, and 6.69 (SD = 1.36) for E group”. 

Also, if students are nested in classrooms, which are then nested in schools, and then neighborhoods, a study on intergroup contact might want to contemplate multilevel modeling. Was this data collected (i.e., who is in what classroom/School)? The analysis might benefit from “who am I in contact with in this classroom /school” in ADDITION to the self-reported “Quality of contact”.

We have the school and grade level data for each student (not the exact classroom). However, one of the aims of this study was the implementation of psychological network analysis. We know and agree with the reviewer in regards to the usefulness of using multilevel analyses, but using these analyses would imply approaching this study from a completely different methodological perspective than the proposed objective. We agree with the reviewer that "the analysis might benefit from who am I in contact with in this classroom /school, in addition to the self-reported quality of contact”. But the research design did not include the collection of these data (only quality of contact). However, we have included a comment on this in the discussion section, pointing out limitations and future lines of research. 

Other, more minor questions or considerations

If students did not identify as Spanish, Moroccan, Romanian or Ecuadorian did they not fill out a questionnaire? How were the diferent survey options presented to the students?

All students who were in class and had parental permission filled out the questionnaire. Before the questionnaire was handed out, the adolescents answered two questions (in paper and pencil format) about their origin (country of birth and country of birth of their parents, both mother and father). In this way, the researchers determined their origin. If they were Spanish adolescents (from Spanish parents) or from any of the origins included in this study (Moroccan, Rumanian or Ecuadorian), they were administered the corresponding questionnaire. If they were from any other ethnocultural background, we administered a different questionnaire that asked in a general way about the country of origin of their parents (taking into account that many of them could be second generation). An example of a question in this general questionnaire was: “To what degree do you currently maintain the customs of your parents’ country of origin?” The data from the participants who responded to this general questionnaire have not been included in the present study. We have not included this information in the manuscript so as not to extend the length of the manuscript excessively. However, if the reviewer considers it appropriate, we could briefly add it in a footnote in Procedure section. 

REVIEWER #2:

The manuscript investigates the perception of different ethnic minorities in light of the SCM and Bias Map. The manuscript is well written and easy to follow. The study considers un understudied sample (adolescents) and employed an innovative approach (network analysis). Therefore, I am happy to recommend this paper for publication after a revision.

1) The distinction between sociability and morality has been dismissed lightly. I would suggest the authors better describe the innovative value of their theoretical approach. They could describe the moral primacy model of impression development (Brambilla et al., 2021; Goodwin, 2015) and state how the present study fits/complements/extends prior insights on the distinction between sociability and morality.

Brambilla, M., Sacchi, S., Rusconi, P., Goodwin, G. (2021). The primacy of morality in impression development: Theory, research, and future directions. Advances in Experimental Social Psychology, 64, 187-262.

Goodwin, G. P. (2015). Moral character in person perception. Current Directions in Psychological Science, 24, 38-44.

Some information on the Moral Primacy Model has been added in the version of the manuscript (lines 86-92): “Some of the above findings have been captured in the three key assumptions of the Moral Primacy Model [25]: 1) Morality, sociability, and competence are distinct dimensions that make unique contributions to impression formation; 2) Morality has a primary role in the formation of impressions and the evaluations we make about other people and groups both at different stages of impression formation and in behavioral outcomes; 3) The primary role of morality is due to its close link to the judgment of whether other social targets represent an opportunity or a threat.”

The following information has also been included in the Discussion section (lines 402-404): “Despite this, the present study is in line with previous findings on the distinction between sociability and morality and covered in the Moral Primacy Model [25].”

2) Hypotheses: The distinction between morality and immorality is well explained in the general discussion. However, I would suggest describing such a distinction only in the introduction. This would be functional to better understand the hypotheses. I would refer to prior work on the confirmability of traits-concepts (Reeder & Brewer, 1979).

Reeder, G. D., & Brewer, M. B. (1979). A schematic model of dispositional attribution in interpersonal perception. Psychological Review, 86, 61.

Part of the information that appeared in the Discussion section has been moved to the Introduction section and the reference to the article recommended has been included (lines 58-61): “Previous research [13-17] has shown evidence of the theoretical importance and the diagnostic capacity of the negative pole of the morality domain (i.e., immorality) in the out-group’s impression formation (i.e., for this dimension, negative information is more enlightening than positive information).”

3) I do not immediately understand why the authors measured the negative and the positive pole of morality, while sociability and competence were measured by considering only the positive pole. Please explain.

Some studies (e.g., Goodwin & Darley, 2012; Sayans-Jiménez et al., 2017) have shown the importance of also considering negative aspects of morality (i.e., immorality) in impression formation. This has not currently been shown with the sociability and competence dimensions. Moreover, the study by Sayans-Jiménez et al. (2017) showed that adding negative items to the sociability and competence dimensions worsened the psychometric properties of a scale for measuring stereotype content. The decision to consider morality and immorality dimensions as separate dimensions is supported by previous studies such as Pittinsky et al. (2011) or Cacioppo & Berntson (1994) and those mentioned in the manuscript.

4) I would also suggest describing more in detail how morality has been conceptualized. Morality is a broad construct defined by distinct components (see for instance the distinction between deontology and consequentialism - Sacchi et al., 2014; or the distinction between harm and fairness, Gray et al., 2009).

Sacchi, S., Riva, P., Brambilla, M., & Grasso, M. (2014). Moral reasoning and climate change mitigation: The deontological reaction toward the market-based approach. Journal of Environmental Psychology, 38, 252-261.

Gray, K., & Wegner, D. M. (2009). Moral typecasting: divergent perceptions of moral agents and moral patients. Journal of personality and social psychology, 96(3), 505.

Thank you very much for the recommendation of these interesting papers. Because the conceptualization of morality is so well established in the psychosocial literature on the stereotype content and models collected in our study (and so as not to overextend the length of the paper) we have decided not to add this information to the manuscript. 

We sincerely thank the reviewers for their valuable comments and recommendations, which have undoubtedly contributed to improving and clarifying our article.

 

REVIEWER #3: 

This is a very interesting paper that contributes to the intergroup attitudes literature. It is well-performed and results are clearly exposed. As for suggestions that could improve the paper are: 

Including in the abstract the application of the results

The following information has been added to the abstract (lines 32-37): “This could indicate that, to achieve successful intergroup relations involving cooperation and the development of friendly relationships, it would be appropriate to intervene in parallel in these two pathways. Due to the centrality of positive emotions (and sociability and immorality) and, therefore, their capacity to affect the entire network, focusing interventions on these variables could be an appropriate strategy to achieve overall positive attitudes.”

Deepen on the conclusions as for the implications of the results

Thank you very much for the suggestion. The implications of the study have been extended to the role of the quality of intergroup contact (lines 461-467): “In this study, it is observed that intergroup contact quality plays a key role in improving intergroup attitudes. The direct relationship between the quality of contact and positive emotions (one of the most central variables of the network) indicates that achieving quality relationships could have an important effect on the whole attitudinal network.”

Performing further analyses to guarantee the comparability of first and second-generation of the group members. 

We did not have an ample enough sample to divide adolescents of immigrant origin into first and second generation. However, following the reviewer's suggestion, we carried out t-tests, comparing the mean scores obtained by both generations in all the study variables. The results of these analyses can be seen below. We found that there were no statistically significant differences in practically any of the variables included in the study, so we decided to use both generations together.

The results of the t-tests for each of the groups can be seen in the attached file titled Response to Reviewers.

Moreover, were there any difference by different schools? maybe the incomes, sex or other variables should also be considered.

Thank you very much for the suggestion. Our objective was to compare these empirical network models in different groups of adolescents as a function of their ethnocultural group (or the ethnocultural group that they evaluate). Therefore, our data collection was aimed at this objective and we do not have an ample enough sample to compare based on the variables that you propose. Despite this, and regarding the specific variables you mention.

We do not have a measure of income, but we do have a measure of perceived socioeconomic status. It can be seen from the responses to reviewer #1 that there are hardly any differences in perceived status between the different groups and that this variable does not produce important changes in the networks. 

Regarding sex, and despite the fact that, as we have mentioned, we do not have an ample enough sample to make these comparisons nor was it our objective, we explain that the percentage of boys and girls is more or less balanced in all groups (around 50%).

Thanks again for your suggestions. We believe that they contribute substantially to improve the manuscript. 

Best regards

The authors

---

## [Decision Letter · Decision Letter 1]

1 Jun 2022

PONE-D-21-14451R1Intergroup attitudes and contact between Spanish and immigrant-background adolescents using network analysis.PLOS ONE

Dear Dr. Sánchez Castelló,

Thank you for submitting your manuscript to PLOS ONE. After careful consideration, we feel that it has merit but does not fully meet PLOS ONE’s publication criteria as it currently stands. Therefore, we invite you to submit a revised version of the manuscript that addresses the points raised during the review process.

Reviewer 1 felt that the manuscript has improved greatly, however, s/he is not able to assess whether the data analysis is conducted to the satisfactory level. I also am not sure how your results address the hypotheses. Please assist the readers in revising the part of the results. That is, how are the current results respond to the hypotheses listed on p. 9. Please refrain from using abbreviations, it makes readers very difficult to follow the results. 

We look forward to receiving your revised manuscript.

Kind regards,

I-Ching Lee

Academic Editor

PLOS ONE

Journal Requirements:

Reviewers' comments:

Reviewer's Responses to Questions

**Comments to the Author**

1. If the authors have adequately addressed your comments raised in a previous round of review and you feel that this manuscript is now acceptable for publication, you may indicate that here to bypass the “Comments to the Author” section, enter your conflict of interest statement in the “Confidential to Editor” section, and submit your "Accept" recommendation.

Reviewer #1: All comments have been addressed

Reviewer #3: All comments have been addressed

2. Is the manuscript technically sound, and do the data support the conclusions?

Reviewer #1: Yes

Reviewer #3: Yes

3. Has the statistical analysis been performed appropriately and rigorously? 

Reviewer #1: I Don't Know

Reviewer #3: Yes

4. Have the authors made all data underlying the findings in their manuscript fully available?

Reviewer #1: Yes

Reviewer #3: No

5. Is the manuscript presented in an intelligible fashion and written in standard English?

Reviewer #1: Yes

Reviewer #3: Yes

6. Review Comments to the Author

Reviewer #1: I see that most of my comments have been addressed, albeit, in some instances quite superficially. Nevertheless, I believe the article has been improved satisfactorially.

Reviewer #3: Thank for addressing all the suggestions and improve the paper. I feel it is suitable for publication in its present form.

7. PLOS authors have the option to publish the peer review history of their article (what does this mean?). If published, this will include your full peer review and any attached files.

Reviewer #1: No

Reviewer #3: **Yes: **Esther Lopez-Zafra

---

## [Author Response · Author response to Decision Letter 1]

20 Jun 2022

Dear Editor and Reviewers,

Thank you very much for your comments. Your recommendations have been included in the new version. The changes are explained below.

Reviewer 1 felt that the manuscript has improved greatly, however, s/he is not able to assess whether the data analysis is conducted to the satisfactory level. I also am not sure how your results address the hypotheses. Please assist the readers in revising the part of the results. That is, how are the current results respond to the hypotheses listed on p. 9. Please refrain from using abbreviations, it makes readers very difficult to follow the results. 

We have made some changes in the results section by linking each result to the corresponding hypothesis and eliminating all abbreviations. 

In addition, the following clarification has been added regarding the structure of the networks (lines 353-358): 

The nodes corresponding to the stereotypes appear as a structure at the top of the networks in all samples. Similarly, nodes corresponding to positive and negative emotions appear below the stereotype structure. At the bottom of the networks, facilitation and harm nodes appear at both ends and are related to emotions. Finally, the contact quality node appears, also in the lower part of the network, and, in general, related to the facilitation and positive emotions nodes.

Abbreviations have also been removed in the discussion section. 

We hope that this will make the reading of the results clearer and simpler (all these changes have been introduced with change control in the manuscript).

Journal Requirements:

We have checked the list of references to ensure that it is complete and correct. We have no citations for papers that have been retracted.

---

## [Editor Report · Decision Letter 2]

30 Jun 2022

Intergroup attitudes and contact between Spanish and immigrant-background adolescents using network analysis.

PONE-D-21-14451R2

Dear Dr. Sánchez Castelló,

We’re pleased to inform you that your manuscript has been judged scientifically suitable for publication and will be formally accepted for publication once it meets all outstanding technical requirements.

Kind regards,

I-Ching Lee

Academic Editor

PLOS ONE
---

## [Editor Report · Acceptance letter]

14 Jul 2022

PONE-D-21-14451R2 

Intergroup attitudes and contact between Spanish and immigrant-background adolescents using network analysis 

Dear Dr. Sánchez-Castelló:

I'm pleased to inform you that your manuscript has been deemed suitable for publication in PLOS ONE. Congratulations! Your manuscript is now with our production department. 

Kind regards, 

on behalf of

Dr. I-Ching Lee 

Academic Editor

PLOS ONE